# Antimicrobial and Antibiofilm N-acetyl-L-cysteine Grafted Siloxane Polymers with Potential for Use in Water Systems

**DOI:** 10.3390/ijms20082011

**Published:** 2019-04-24

**Authors:** Dorota Kregiel, Anna Rygala, Beata Kolesinska, Maria Nowacka, Agata S. Herc, Anna Kowalewska

**Affiliations:** 1Institute of Fermentation Technology and Microbiology, Lodz University of Technology, Wolczanska 171/173, 90-924 Lodz, Poland; 2Institute of Organic Chemistry, Faculty of Chemistry, Lodz University of Technology, Zeromskiego 116, 90-924 Lodz, Poland; beata.kolesinska@p.lodz.pl; 3Centre of Molecular and Macromolecular Studies, Polish Academy of Sciences, Sienkiewicza 112, 90-363 Lodz, Poland; mnowacka@cbmm.lodz.pl (M.N.); asherc@cbmm.lodz.pl (A.S.H.); anko@cbmm.lodz.pl (A.K.)

**Keywords:** bacteria, biofilm, N-acetyl-L-cysteine, polysilsesquioxanes, polysiloxanes

## Abstract

Antibiofilm strategies may be based on the prevention of initial bacterial adhesion, the inhibition of biofilm maturation or biofilm eradication. N-acetyl-L-cysteine (NAC), widely used in medical treatments, offers an interesting approach to biofilm destruction. However, many Eubacteria strains are able to enzymatically decompose the NAC molecule. This is the first report on the action of two hybrid materials, NAC-Si-1 and NAC-Si-2, against bacteria isolated from a water environment: *Agrobacterium tumefaciens*, *Aeromonas hydrophila*, *Citrobacter freundii*, *Enterobacter soli*, *Janthinobacterium lividum* and *Stenotrophomonas maltophilia*. The NAC was grafted onto functional siloxane polymers to reduce its availability to bacterial enzymes. The results confirm the bioactivity of NAC. However, the final effect of its action was environment- and strain-dependent. Moreover, all the tested bacterial strains showed the ability to degrade NAC by various metabolic routes. The NAC polymers were less effective bacterial inhibitors than NAC, but more effective at eradicating mature bacterial biofilms.

## 1. Introduction

When biofilm communities develop on industrial surfaces, they constitute a reservoir of various bacterial strains, including pathogens and opportunistic pathogens. The formation of biofilms often results in difficult-to-treat, chronic infections. In the case of technical materials in drinking water systems, they can lead to secondary contamination of water and biocorrosion processes [1,2]. Moreover, adhered bacterial cells produce extracellular polymeric substances (EPSs) which promote the development of biofilm structures. Bacterial biofilms usually express starvation phenotypes and defense mechanisms. As a consequence, they become more resistant to biocids than planktonic cells [3].

Potential antibiofilm strategies may be based on prevention of initial bacterial adhesion, inhibition of biofilm maturation and biofilm dispersion or eradication [4]. One common approach is to treat or incorporate surfaces with biocides, such as metal ions or metal nanoparticles (silver, cooper and mercury) or chemical agents (triclosan, chlorhexidine and quarternary ammonium salts). However, these approaches are limited, in particular by the short lifetimes and toxicity of such biocides [5].

An interesting alternative antibiofilm strategy may be the use of N-acetyl-L-cysteine (NAC), which is widely used in medical treatments. This small molecule shows rather weak antibacterial properties, but NAC is well known as a strong biofilm inhibitor. As a thiol antioxidant, NAC disrupts disulfide bonds in EPS as well as inhibiting cysteine utilization. In addition, NAC increases the wettability of the surface and decreases bacterial adhesion to substrates [3]. It has been shown that NAC can reduce biofilms formed by some bacteria: *Staphylococcus epidermidis*, *Stenotrophomonas maltophilia* and *Burkholderia cepacia* [6,7], inhibit biofouling by *Moraxella catarrhalis*, *Streptococcus pneumoniae* and *Haemophilus influenzae* [8,9] and limit the adhesion of *Pseudomonas aeruginosa* to polystyrene [10] and of *Helicobacter pylori* to various biotic and abiotic surfaces [11,12].

The main barrier to widespread use of NAC as a universal antibiofilm agent may be the ability of many Eubacteria strains to decompose this molecule via the action of various microbial enzymes [13,14,15,16]. Especially in water environments, where the concentration of organic matter is very low, NAC becomes an attractive substrate for bacteria. Regardless of the mechanism of action, the antibiofilm activity of NAC may be significantly reduced. Therefore, in many cases, antibiofilm treatment by NAC should be supported by the additional activity of antibiotics, silver and gold nanoparticles or other antimicrobials [17,18,19,20,21]. Another strategy could be the incorporation of NAC into the structure of larger macromolecules, thereby limiting enzymatic degradation. So far, there has been little research on functional polymeric materials containing NAC or their effects on bacterial biofilms. Interesting results have been reported for chitosan [22] and polysiloxanes [23] functionalized with NAC. However, in these studies, bacterial ATTC collection strains of *Staphylococcus aureus* and *Escherichia coli* were used as model microorganisms. This paper marks an extension of studies into NAC grafted onto functional siloxane polymers using bacterial strains isolated from biofilms formed in industrial drinking water systems. This is the first report on NAC activity against wild bacterial isolates and the application of NAC siloxane polymers against mature bacterial biofilms.

## 2. Results

### 2.1. Bacterial Isolates

In total, 40 bacterial monocultures were isolated from the biofilm samples, using the plate reduction technique on PCA agar. The bacterial morphotypes varied, with beige, cream, colorless or dark-violet colonies of regularly shaped small (2 mm) and irregular larger (4–5 mm) colonies, all with a characteristic slimy appearance. In general, the morphotypes were Gram-negative bacteria. The mixed bacterial populations were very difficult to separate. Therefore, the streak plate procedure was performed several times on PCA agar to obtain each morphotype as a pure culture. Finally, six bacterial morphotypes with the ability to form characteristic slimy colonies on agar plates were selected for genetic identification. The bacterial monocultures were identified at the species level by sequencing their 16S rRNA genes, which were amplified through PCR. The nucleotide sequences were compared with those obtained from the National Center for Biotechnology Information (NCBI) and deposited in the GenBank database with accession numbers. The isolated bacterial strains were identified as *Agrobacterium tumefaciens*, *Aeromonas hydrophila*, *Citrobacter freundii*, *Enterobacter soli*, *Janthinobacterium lividum* and *Stenotrophomonas maltophilia* (Table 1).

Some of these isolates were typical water microbiota. According to the literature, Gram-negative bacteria are commonly found in water systems. Studies conducted by Penna et al. [24] on water samples collected from public distribution installations confirmed the presence of the Gram-negative bacteria *Pseudomonas* sp., *Flavobacterium* sp., *Acinetobacter* sp. and *Enterobacteriaceae*, *C. freundii* and *E. soli* (coliforms) are opportunistic pathogens often found in water, sewage and the intestinal tracts of animals and humans [25,26,27]. *A. tumefaciens* originates in plants and soil [28,29]. *S. maltophilia* frequently colonizes humid abiotic surfaces in water installations, mechanical ventilation systems and medical devices [30], as do *A. hydrophila* [31,32,33] and *J. lividum*, which are often isolated from water and water-living animals [34]. *S. maltophilia* is closely related to *Pseudomonas* and *Xanthomonas* genera. They are opportunistic pathogens that cause human respiratory tract infections, endocarditis, bacteremia, meningitis and urinary tract infections, which are often difficult to treat [35]. Despite their different taxonomic characteristics and biochemical features, the bacterial isolates were able to form slime on the PCA agar plates, often in colonies with compact structures. It seems that this feature helps them to adhere to or coaggregate with other cells and surfaces, while strengthening the structures of the biofilms.

### 2.2. Growth Inhibition of Bacteria by NAC

Six bacterial strains, *A. tumefaciens*, *A. hydrophila*, *C. freundii*, *E. soli*, *J. lividum* and *S. maltophilia*, were used to evaluate the antibacterial activity of NAC. The influence of this compound was determined by the standard two-fold dilution method in two kinds of culture media: minimal M1 (50-fold diluted buffered peptone water, BPW) and rich M2 (Triptic Soy Broth, TSB). No significant reduction in bacterial density was observed after 48 h of incubation for any of the strains in M2 medium, except *A. tumefaciens*. The NAC concentrations at which bacterial growth [°McF] was slightly inhibited were similar for all the studied strains, at 0.25% (*w*/*v*). However, at this concentration, the inhibition of bacterial growth by NAC was significantly higher in the minimal M1 medium (Figure 1). These results show that the antibacterial activity of NAC depends on the availability of organic compounds which in effect can protect planktonic bacterial cells.

Numerous studies have confirmed that organic matter interferes with the activity of antimicrobials [36,37,38]. The interaction between organic matter and antimicrobial substance leads to reduced antimicrobial activity. Therefore, the antimicrobial activity of NAC can be obscured by the environment and depends strongly on its chemical composition. Similarly, Yin et al. [39] suggest that NAC may be unsuitable as an antibacterial agent in the presence of high concentrations of organic matter.

### 2.3. Adhesion Abilities in the Presence of NAC

Bacterial attachment to the native glass surface in M1 culture medium was assessed by luminometry throughout the 6-day incubation period. This minimal medium was used because, according to the literature, both biofilm formation and biofilm resistance to antimicrobials may be stimulated in a water environment poor in carbon sources. Myszka and Czaczyk [40] report that starvation has a greater impact on the process of cell attachment. Other studies conducted by Rochex and Lebeaultthe [41] have shown that the extent of biofilm accumulation increases as the nitrogen concentration falls from C/N = 90 to C/N = 20. In our study, for culture media M1 and M2, the C/N ratios were likewise very low, at 3.63/3.95. However, in the M1 medium, the availability of organic compounds was very limited (0.2 g/L) (Table 2).

Figure 2 presents the results of luminescence studies in Relative Light Units per cm^2^ (RLU/cm^2^) for the tested bacterial strains. Various levels of cell adhesion may be observed. For *A. hydrophila*, *C. freundii*, *J. lividum* and *S. maltophilia*, the RLU values in the control samples are higher, at 121–143 RLU/cm^2^. The presence of 0.25% (*w*/*v*) NAC significantly reduced the bacterial adhesion of *J. lividum* and *S. maltophilia*, to 2–3 RLU/cm^2^. For other strains, the RLU values increased from 26 to 56 RLU/cm^2^.

According to the literature, NAC can reduce the formation of biofilms by *S. epidermidis* [6], *P. aeruginosa* [10], *H. pylori* [11], *S. maltophilia* and *B. cepacia* [7] on various biotic and abiotic surfaces. Biofilms have been reported to be more affected by NAC than planktonic cultures, suggesting specific antibiofilm activity against the tested bacteria [7]. The results of our study confirmed the antibiofilm action of NAC. However, the final effect was strain-dependent. The variable activity of NAC may be due to the chemical nature of the EPS, as well as to the enzymatic activity of the bacterial strains capable of NAC degradation. The chemical characteristics of bacterial EPS are strain-dependent, and also depend on the age of the biofilms [42]. Therefore, it was decided to evaluate the capacity of the tested bacteria to degrade NAC.

### 2.4. Bacterial Capacity for NAC Degradation

A comparison of the HPLC chromatograms and MS spectra for solutions obtained after the incubation of bacterial cells with the control NAC solution reveals that all the tested bacterial strains showed ability to degrade NAC in water, when other potential sources C and N were absent. Figure 3A presents examples of chromatograms for three tested bacterial strains: *A. hydrophila*, *C. freundii* and *E. soli*. These chromatograms have slight differences from each other, but present definite differences from that of the control sample.

The LC–MS spectra for all the analyzed supernatants show a peak at the retention time of 9.1 min, for which *m*/*z* is 325.1. This signal corresponds to the [M + H]^+^ product of NAC oxidation to the disulfide derivative. More polar compounds were found in the supernatants in comparison to those in the NAC solution without bacterial incubation (Figure 3B). Small amounts of alanine (visible as a peak at *m*/*z* = 90.5 on the MS spectra, corresponding to [M + H]^+^ alanine) were present in the supernatants. Surprisingly, the supernatants did not contain cysteine, which is the product of deacetylation of NAC. The peak retention time of *A. tumefaciens* was 2.7 min (*m*/*z* = 283.1) (Figure 3C). The *m*/*z* corresponds to the disulfide derivative formed from the NAC and cysteine. A compound of *m*/*z* = 118.1 with a retention time of 2.4 min was observed among the metabolites of *E. soli*. This was tentatively assigned to N,N-dimethylalanine (Figure 3D).

Several enzymes may be responsible for the degradation of cysteine to different metabolites, namely cysteine desulfurase (CDS) (EC 2.8.1.7) [13,16], cysteine desulfhydrase (CDSH) (EC 4.4.1.15) [15] or cysteine dioxygenase (CDO) (EC 1.13.11.20) [14]. For example, CDS and CDSH activities have been found in *Salmonella enterica* and *E. coli*, and CDO is common among species within the phyla of *Actinobacteria*, *Firmicutes* and *Proteobacteria*. These enzymes differ in terms of action: CDO irreversibly oxidizes the sulfhydryl group of cysteine to cysteinesulfinic acid, whereas CDS and CDSH, which appear to be the major cysteine-degrading agents, are sulfide-producing enzymes [14,15].

Due to the ability of the tested bacterial strains to degrade NAC, subsequent studies were conducted on NAC polymer derivatives with different structures: ladder-like (NAC-Si-1) and linear (NAC-Si-2). Two bacterial strains were used, *A. hydrophila* and *C. freundii*. These exhibited the best adhesive abilities, comparable resistance to NAC as well as a capacity for NAC degradation.

### 2.5. Growth Inhibition by NAC and NAC-Grafted Polymers

The antibacterial activities of NAC and of the polymers NAC-Si-1 and NAC-Si-2 were evaluated in two kinds of culture media: minimal M1 and enriched M2 (Figure 4).

Aqueous solutions of NAC and NAC-Si-1 were used at concentrations of 0.25% (*w*/*v*). Due to its limited solubility, NAC-Si-2 was applied at 0.05% (*w*/*v*). After 48 h incubation in minimal M1 medium, a significant reduction in bacterial growth was observed only in the case of native NAC. In enriched M2 medium, the strains were inhibited only slightly in the presence of NAC and its derivatives. Growth inhibition was similar for both tested strains. These results show that the wild-bacterial isolates are more resistant to NAC, NAC-Si-1 and NAC-Si-2 in M2 medium than the model collection strains *E. coli* and *S. aureus* described in our previous report [23]. Nevertheless, in M2 medium, the activity of NAC-Si-2 was similar to that of NAC-Si-1 at a 5-fold lower concentration.

### 2.6. Biofilm Formation on Glass Modified with NAC Polymers

A significant reduction in bacterial biofilm formation after 6 days of incubation in minimal M1 medium was observed only in the case of polymer NAC-Si-2. A similar, 10-fold reduction was observed with both tested strains (Figure 5).

The NAC-Si-1 polymer proved less effective at reducing bacterial adhesion. This may be due to the differences in wettability of the polymers. Numerous publications have investigated the relationships between the surface properties of materials and biofilm colonization, although some details remain unclear. Bacterial adhesion depends not only on the bacterial strain but also on the surface free energy of the colonized support. Generally, larger specific surface areas and better wetting qualities have been found to favor bacterial adhesion [43]. Therefore, evaluating polymer wettability can give important information regarding the antibiofilm properties of materials.

The surface free energies of thin films of NAC-Si-1 and NAC-Si-2 on glass were compared to that of native glass. The surfaces covered with polymers were more hydrophilic than that of the native glass support, which had a surface free energy of 180 mJ/m^2^, in comparison to 240 mJ/m^2^ and 380 mJ/m^2^ for NAC-Si-1 and NAC-Si-2, respectively. The specific structure of NAC-Si-2 is responsible for its particular properties in thin films and great hydrophilicity, despite the presence of hydrophobic methyl groups on Si atoms. Moreover, the solubility of NAC-Si-2 in water is poorer than that of NAC-Si-1. As we have seen, the greater flexibility of the single siloxane chain of NAC-Si-2 facilitates the rearrangement of macromolecules in the coating, so the hydrophilic groups in the topmost layer of the film may be exposed to the polar environment. In our previous studies, it was shown that NAC-Si-2 ‘catches and holds’ bacterial cells, and this can impede biofilm development [23].

### 2.7. Biofilm Eradication by NAC and NAC-Grafted Polymers

The ability of the tested polymers to eradicate biofilms was equally interesting. We studied eradication using different kinds of biofilm treatment: distilled water, 1% (*w*/*v*) solutions (NAC, NAC-Si-1) and suspensions (NAC-Si-2) (Figure 6). It was noted that the treatment of bacterial biofilms with NAC or its polymers at a concentration of 1% (*w*/*v*) led to a significant decrease in RLU values compared to the control samples. However, the application of NAC-Si-2 polymer seems to be the best approach to eradication [23].

However, microscopic observations (Figure 7) suggest that the process of eradication is based on interactions at the interface between the biofilm and the macromolecules. Analysis using SEM of the glass surface following eradication processes showed that the less water-soluble polymer, NAC-Si-2, has a tendency to ‘clump and hold’ bacterial cells, which may result in very low luminometric measurements (due to lack of cell availability for intracellular ATP testing). Moreover, this effect was probably the cause of the lack of cell fluorescence when we attempted to determine the metabolic state of the bacterial cells by fluorescence microscopy. Groups of tightly packed bacterial cells were also noticed (Figure 7B-4). This particular feature of the polysiloxane explains the lack of fluorescence emission from bacterial cells eradicated with NAC-Si-2. Therefore, it can be concluded that the polymer NAC-Si-1, with better water solubility, is more effective against mature bacterial biofilms.

## 3. Materials and Methods

### 3.1. Isolation of Bacterial Biofilms Formed in Drinking Water Systems

In total, 20 samples of biofilms formed in industrial drinking water systems were subjected to microbiological analysis. For each biofilm sample, at least three plates with Plate Count Agar (PCA, Merck, Darmstadt, Germany) were inoculated by swabbing. All the plates were incubated at 25 °C for 5 days. At least two characteristic colonies representing each morphotype were picked up from the agar plates, restreaked several times to ensure purity and then maintained as pure cultures at 4 °C on wort agar slants.

### 3.2. Identification of Microorganisms

The bacterial cultures were analyzed first by light microscopy BX41 (Olympus, WA, USA), and then by using molecular methods and the following standard methods: Gram staining, the L-aminopeptidase test (Bactident^®^ Aminopeptidase, Merck, Germany), the oxidase test (Bactident^®^ Oxidase, Merck, Germany). The 16S rRNA genes were amplified through PCR, according to a technique described previously [44]. The nucleotide sequences were compared with 16S rRNA gene sequences obtained from the NCBI using the program BLASTN 2.2.27 + (https://blast. ncbi.nlm.nih.gov/Blast.cgi). The sequences were deposited in the GenBank database and assigned accession numbers.

### 3.3. Culture Media

Certified liquid media were prepared for the shake cultures. The minimal medium M1 (50-fold diluted BPW (Merck, Germany)) and enriched medium M2 (TSB (Merck, Germany)) were sterilized at 121 °C for 15 min. These media were chosen due to their sharp differences in nutrient content. The composition of the granulates used to prepare the M1 and M2 media was determined by elemental analysis (%wt of C, N and S) using a varioMICRO CUBE elemental analyzer (Elementar, Langenselbold, Germany). Table 2 presents the chemical characteristics of the culture broths used. The values are the means of two independent measurements (*n* = 2).

### 3.4. Polymeric Materials

Polymers, labelled NAC-Si-1 and NAC-Si-2, were prepared by grafting NAC onto the vinyl groups of their precursors, LPSQ-Vi and Silo-Vi, respectively, following a method described elsewhere [23,45,46]. The polymers with side vinyl groups were poly(vinylsilsequioxanes) (LPSQ-Vi, MnGPC = 1000 g/mol, PDI = 1.4) or poly(vinylmethylsiloxanes) (Silox-Vi, Gelest, MnGPC = 1800 g/mol, PDI = 1.3) terminated with trimethylsilyl groups (Figure 8). To study biofilm growth on surfaces coated with NAC grafted onto siloxane polymers, the surface energy of the polymers was estimated using the sessile droplet technique and the Owens–Wendt method. The polymers were cast on bare glass supports (commercially available microscopic slides) using a slit applicator (film thickness: 100 μm). The static, advancing and receding contact angles were measured immediately after the deposition of a liquid (water or anhydrous glycerol) onto the surface of the film. The values and their standard deviations were estimated for the average of at least three measurements taken in different areas of the same sample [23].

### 3.5. Bacterial Growth in the Presence of NAC or NAC-Grafted Polymers

The influence of NAC and its derivatives on the tested bacteria were determined using the standard two-fold dilution method [23]. One milliliter of cell suspension (1°McF) was mixed with 1 mL of M1 or M2 culture media with serial dilutions of the tested compounds. The concentrations of NAC (Merck, Germany) and its polymeric derivatives NAC-Si-1 and NAC-Si-2 (Figure 8) were in the range of 0.0125–0.5% (*w*/*v*). Incubation was conducted for 48 h at 30 °C. Bacterial growth in the two types of culture media, M1 and M2, was measured densitometrically in McF units (densitometer DEN-1, Merck, Germany).

### 3.6. Capacity of Bacteria to Degrade NAC

The capacity of the tested bacteria to degrade NAC was evaluated after incubation of the bacterial cells with 0.5% (*w*/*v*) NAC solution in water. Two milliliters of NAC solution was mixed with two full loops of bacterial biomass from 24 h slant cultures. The suspensions were standardized (1°McF) and then incubated for 4 h at 30 °C. After incubation, the samples were centrifuged (6800× *g*, 4 °C) and the supernatants were collected. Solutions of the supernatants diluted with water (Milli-Q water) at 1:1 (*v*:*v*) were used for LC–MS analysis. High-performance liquid chromatography (HPLC) was performed on an LC Dionex UltiMate 3000 (ThermoFisher Scientific, Waltham, USA), using a Kinetex Reversed Phase C18 column (100 × 4.6 mm). The analysis was performed with a gradient of 0.1% TFA in H_2_O (B) and 0.1% TFA in CH3CN (A), at a flow rate of 0.4 mL/min, with UV detection at 214, 220, 254 and 330 nm. Microscopic analysis was performed on an MS Bruker microOTOF-QIII (Bruker, Leipzig, Germany).

### 3.7. Bacterial Adhesion in the Presence of NAC or Polymers-Grafted with NAC

Minimal medium M1 (20 mL) was poured into sterile 25 mL Erlenmeyer flasks, into which sterile glass carriers (Star Frost 76 9 26 mm, Knittel Glass, Braunschweig, Germany) were placed vertically in such a way that half of the carrier was immersed in the medium while the other part remained outside [33]. The amount of inoculum was standardized [1°McF] to obtain a cell concentration in the culture medium approximately equal to 5000–10,000 CFU/mL at the start of each experiment. The samples were incubated at 25 °C on a laboratory shaker (135 rpm) for 6 days. Analysis of the extent of cell adhesion to the glass carriers was performed using luminometry. For luminometric tests, the carrier was removed from the culture medium, rinsed with sterile distilled water and swabbed using free ATP sampling pens (Merck, Germany). The measurements were reported in RLU/cm^2^ using a HY-LiTE 2 luminometer (Merck, Germany) [47,48].

### 3.8. Biofilm Eradication by NAC and NAC-Grafted Polymers

Glass carriers with 10-day old biofilms were rinsed with sterile distilled water and transferred into flasks containing 1% (*w*/*v*) NAC or its polymeric derivatives, NAC-Si-1 or NAC-Si-2. The control sample was transferred directly into sterile water. The bacterial biofilms were then incubated at 25 °C for 1 h using a laboratory shaker at 130 rpm. The glass plates were removed from the incubation emulsions, rinsed with sterile distilled water and swabbed using sterile swabs for surface testing. The number of cells in the biofilm that had formed on the glass surface was determined using luminometry. The results were expressed in RLU/cm^2^ [23]. The surface was also analyzed using scanning electron microscopy (SEM). Images were taken with a JSH 5500 LV scanning electron microscope (JEOL Ltd., Tokyo, Japan) in high-vacuum mode at an accelerated voltage of 10 kV or 15 kV. The samples were splutter-coated with a fine layer of gold, about 20 nm thick, using an ion coating JFC 1200 apparatus (JEOL Ltd., Japan) [49].

### 3.9. Statistical Methods for Biological Samples

Means with standard deviations were calculated from the data obtained from three independent experiments. The mean values of the adhesion results were compared using one-way repeated measures analysis of variance (ANOVA; OriginPro 8.1, OriginLab Corp., Northampton, MA, USA). The results were compared to those for the control samples. Values with different letters presented in the figures show statistically significant differences: a, *p* ≥ 0.05; b, 0.005 < *p* < 0.05; c, *p* < 0.005.

## 4. Conclusions

In this study, NAC was confirmed to have antimicrobial and antibiofilm activity against the Gram-negative strains *A. tumefaciens*, *A. hydrophila*, *C. freundii*, *E. soli*, *J. lividum* and *S. maltophilia*, which are increasingly identified as opportunistic pathogens. Interestingly, these bacterial strains showed the ability to degrade NAC in water, suggesting that the action of NAC may be significantly limited in this environment, due to bacterial enzymatic activities. It is therefore hypothesized that the antibacterial and antibiofilm properties of NAC are multifactorial (i.e., dependent on the bacterial strain and crucial conditions in the environment). New hybrid polymers obtained by grafting NAC onto poly(vinylsilsesquioxanes) and poly(methylvinylsiloxanes) showed rather low antibacterial activity. However, they showed significant ability to eradicate mature biofilms. These novel antibacterial polymers are promising agents for antibiofilm strategies in industrial installations of water.

## Figures and Tables

**Figure 1 ijms-20-02011-f001:**
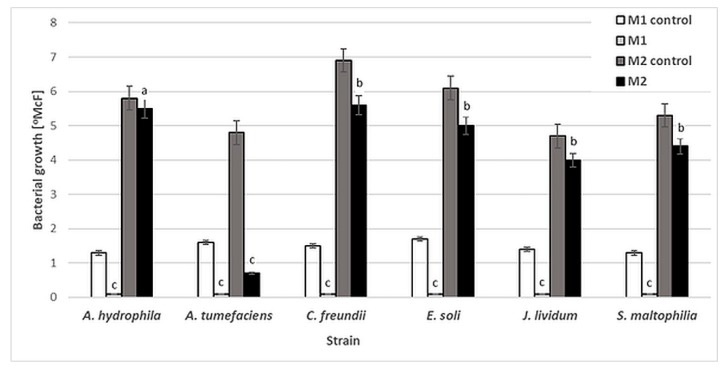
Growth [°McF] of bacterial strains in minimal M1 and rich M2 culture media in the presence of 0.25% (*w*/*v*) NAC in comparison to that for control samples (without NAC). Values show the mean ± standard deviation (SD, *n* = 3). The results are compared to those of the control samples (without NAC). Values with different letters are statistically different: a, *p* ≥ 0.05; b, 0.005 < *p* < 0.05; c, *p* < 0.005.

**Figure 2 ijms-20-02011-f002:**
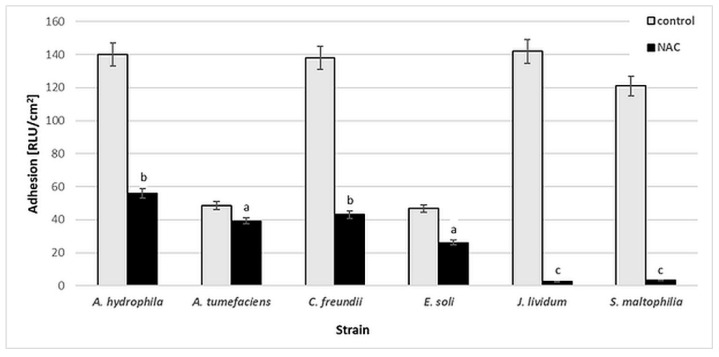
Adhesion [RLU/cm^2^] of bacterial strains to glass surface after 6-day incubation in minimal medium M1 with 0.25% (*w*/*v*) NAC in comparison to that of control samples (without NAC). Values show the mean ± standard deviation (SD, *n* = 3). Values with different letters are statistically different: a, *p* ≥ 0.05; b, 0.005 < *p* < 0.05; c, *p* < 0.005.

**Figure 3 ijms-20-02011-f003:**
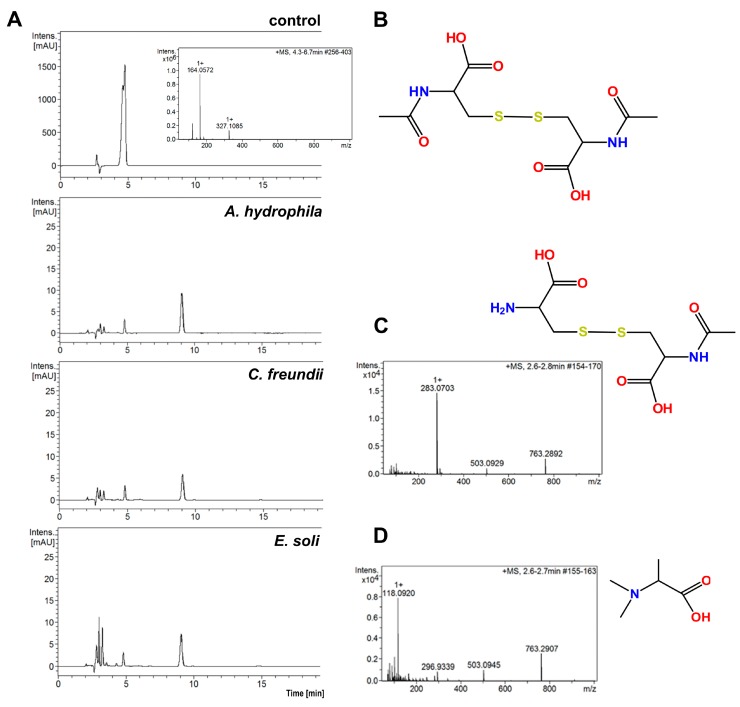
(**A**) HPLC chromatograms of solutions obtained after 4 h of incubation with 0.25% (*w*/*v*) NAC for three bacterial strains. The control was 0.5% NAC solution without bacterial incubation; (**B**) structure of the main NAC metabolite; (**C**) MS spectrum and structure of the polar metabolite of NAC from supernatant after 4 h incubation of *A. tumefaciens* with NAC, (**D**) MS spectrum and structure of polar metabolite of NAC from supernatant after 4 h incubation of *E. soli* with NAC.

**Figure 4 ijms-20-02011-f004:**
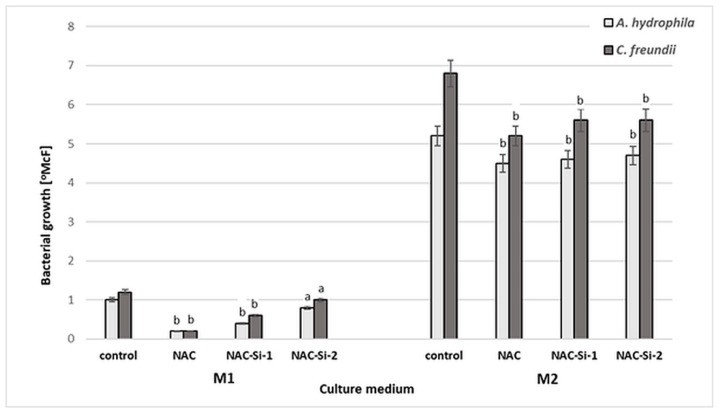
Growth [°McF] of bacterial strains in minimal M1 and enriched M2 culture media in the presence of NAC and its derivatives NAC-Si-1 and NAC-Si-2 in comparison to that in control samples (without NAC and its polymer derivatives). Values show the mean ± standard deviation (SD, *n* = 3). Values with different letters are statistically different: a, *p* ≥ 0.05; b, 0.005 < *p* < 0.05.

**Figure 5 ijms-20-02011-f005:**
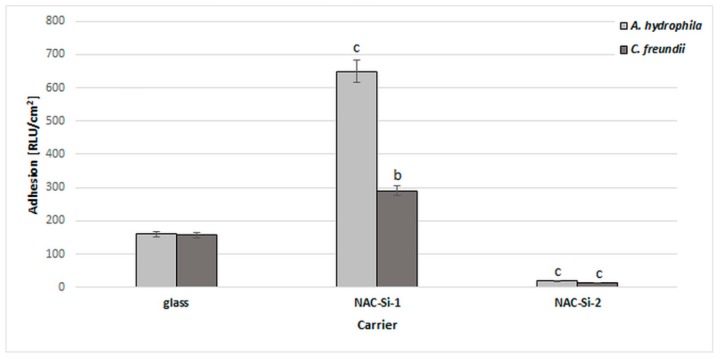
Effects of polymers NAC-Si-1 and NAC-Si-2 on biofilm formation by *A. hydrophila* and *C. freundii* after 6 days of incubation in M1 medium. The results are compared to those of the control glass carrier (glass without NAC polymers). Values show the mean ± standard deviation (SD, *n* = 3). Values with different letters are statistically different: b, 0.005 < *p* < 0.05; c, *p* < 0.005.

**Figure 6 ijms-20-02011-f006:**
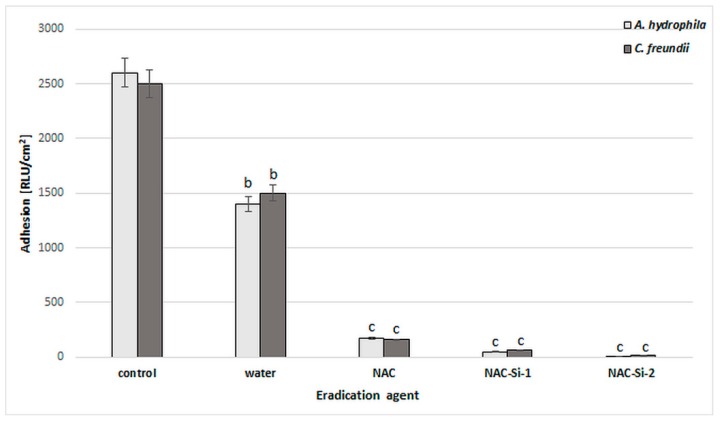
Effects of water, NAC and polymers NAC-Si-1 and NAC-Si-2 on eradication of biofilms formed by *A. hydrophila* and *C. freundii*. Results after 1 h treatment compared to that of the control sample (10-day biofilm without NAC or polymer derivatives). Values show the mean ± standard deviation (SD, *n* = 3). Values with different letters are statistically different: b, 0.005 < *p* < 0.05; c, *p* < 0.005.

**Figure 7 ijms-20-02011-f007:**
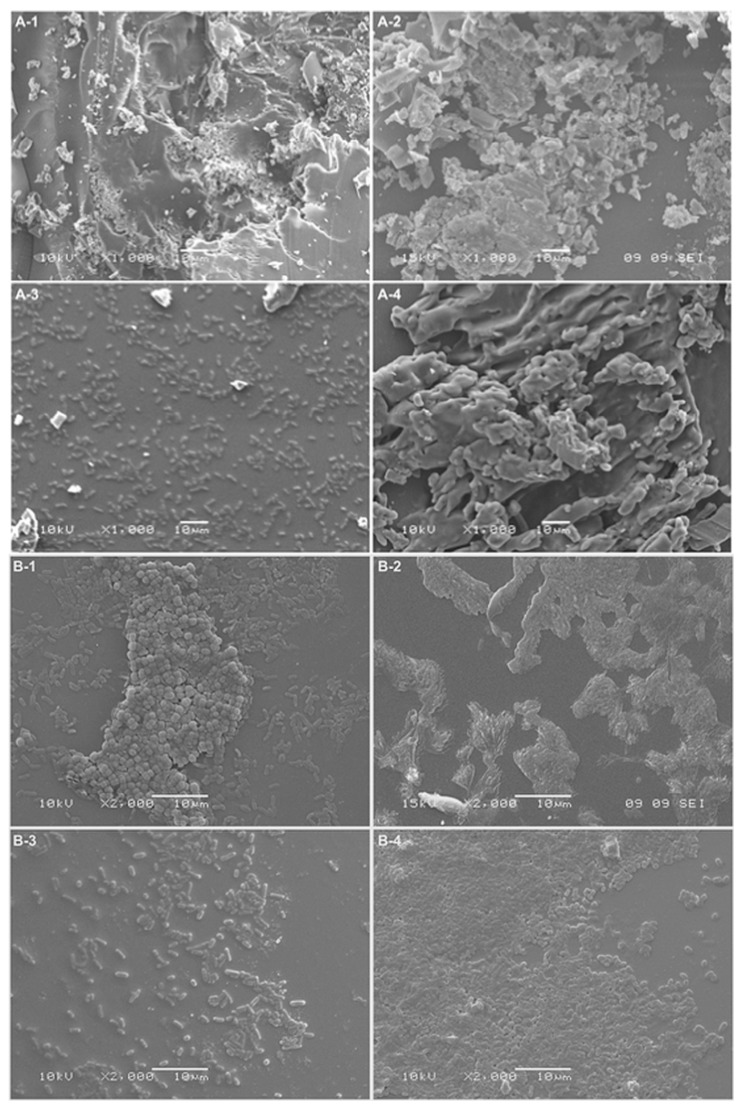
Bacterial attachment observed by SEM: (**A**) *A. hydrophila* biofilms; (**B**) *C. freundii* biofilms; (1) control; (2) NAC eradication; (3) NAC-Si-1 eradication; (4) NAC-Si-2 eradication (scaling bar: 10 μm).

**Figure 8 ijms-20-02011-f008:**
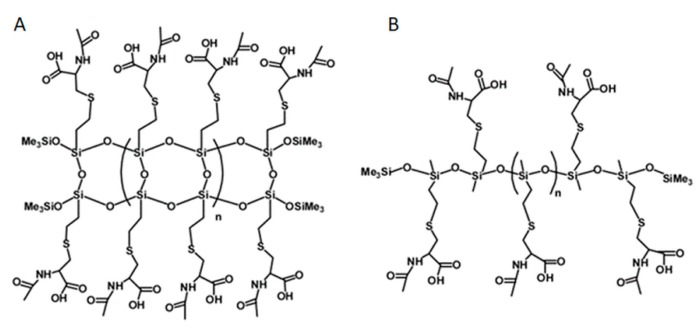
Chemical structures of NAC-Si-1 (**A**) and NAC-Si-2 (**B**) polymers.

**Table 1 ijms-20-02011-t001:** Bacterial strains isolated from biofilms formed in drinking water systems.

Bacterial Isolate	Origin	Cell Shape	Gram Staining	Aminopeptidase L-Alanine	Oxidase	GenBank Accession Number
*A.tumefaciens*	A	rods	negative	positive	positive	KJ719245
*A.hydrophila*	B	rods	negative	positive	positive	KC756842
*C. freundii*	A	rods	negative	positive	negative	KJ995856
*E. soli*	A	rods	negative	positive	negative	KJ995858
*J. lividum*	C	rods	negative	positive	positive	MF777041
*S. maltophilia*	D	rods	negative	positive	weak	KJ719248

A: bottled mineral water; B: unchlorinated drinking water; C: flavored mineral water; D: brewery water.

**Table 2 ijms-20-02011-t002:** Chemical characteristics of culture media used in the study.

Kind of Medium	pH	C [g/L]	N [g/L]	S [g/L]	C/N
M1	6.9	0.08	0.02	0.01	3.63
M2	7.2	8.42	2.13	0.24	3.95

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
