# Peer review of "Antimicrobial and Antibiofilm N-acetyl-L-cysteine Grafted Siloxane Polymers with Potential for Use in Water Systems"

_ijms, 2019, doi:10.3390/ijms20082011_

Round 1
Reviewer 1 Report
The article is well written, the experiment is clear and understandable, the results are well and clearly discussed. Graphical and typographic layout of the text is at a relatively high level, there is only a few minor shortcomings (more in detail below). Despite the clarity of the text and the experiment itself, I have the following remarks on the article and I recommend taking them into consideration:
1) No proof of glass surface condition after modification (except gold coated sample) is present. Picture of no modification, with NAC, NAC-Si-1 and NAC-Si-2 modification is strongly required. At least LM can be done! This data can be presented as supplementary.
2) In addition to MS, to prove changes in the -S-H and -S-S- concentration, XPS is strongly recommended (or FTIR or RA microscopy). Simply stated - this article can be more attractive with some "nice" and "colour" figure(s).
3) Why attached microorganisms (biofilm) on the glass or modified glass was not better visualized (e.g. by the means of some fluorescence technique)?
Other bugs or shortcomings:
Page 5/14, Line 166: I suggest to improve the image, it does not look nice (spectra are cropped/cut). Molecules are probably copied from web. Please pay more attention to this image.
Page 10/14, Line 316 (please apply this comment to the entire manuscript): Do not use RPM but RCF (or RPM and rotor radius).
Page 10/14, Line 316 (please apply this comment to the entire manuscript): the Celsius symbol (°C) looks strange, please make sure the correct symbol is used
Page 338/38, Line 338 (please apply this comment to the entire manuscript): The Celsius symbol (°C) cannot be written as "o" (even with "o" as superscript - it is typographically incorrect)! Please use the correct symbol (you can even find it on keyboard).
The document was checked for plagiarism, some passages show a high degree of compliance (the whole match is 29% - Crossheck, which is not small number). For this reason, it is recommended that authors review the degree of compliance and rephrase at least these passages:
Page 1/14, Line 34, 36, 38-39
Page 2/14, Line 45-46, 74-75, 77-80
Page 3/14, Line 119-121
Page 7/14, Line 240-244
Author Response
Dear Reviewer 1,
We are very grateful for your thorough review of our paper, which has helped us to improve the quality of our publication. We have taken all your remarks into consideration, and have revised our paper in accordance with your comments.
We hope you will find our article after correction suitable for publication in International Journal of Molecular Sciences, section Molecular Microbiology.
Yours faithfully,
Anna Rygala & Dorota Kregiel

Reviewer 2 Report
This manuscript described a method that grafted NAC onto functional siloxane polymers to evaluate its potential on antimicrobial and antibiofilm activities. Although authors demonstrated positive inhibitory effect of NAC polymers on biofilm formation, however, the experimental design and the way data been presented need to be improved.
Line 21
“The NAC was grafted onto functional siloxane polymers to reduce its availability to bacterial enzymes”:
It seems a polymers matrix is an important modification to retain NAC activity. Had authors tried and compared with other polymers?
Line 22
“The results confirm the antibacterial and antibiofilm action of NAC”:
However, on line 43, authors stated NAC shows rather weak antibacterial properties, but is well known as a strong biofilm inhibitor. Here are two questions. First, do these two statements seem to conflict each other, in the perspective of antibacterial activity? Second, if NAC is a well-known strong biofilm inhibitor, how does it been discovered before decomposition by bacteria strains? The novelty of this research seems not high by grafting a well-known antibacterial material onto a matrix. In addition, since the weak antibacterial properties of NAC had been reported, I suggest this manuscript should focus on antibiofilm formation/eradication rather than antibacterial activities.
Line 35
“EPS” needs to be spelled out at the first time.
Line 47
“It has been shown that NAC can reduce biofilms formed by……….. (Ref 6, Ref 7)”
If literatures had reported the biofilm inhibitory effect, authors should clarify the purpose of this research and the need of polymer matrix.
Line 57
“Another strategy could be the incorporation of NAC into the structure of larger macromolecules……. “
Authors need to clarify the approach, small polymer matrix or larger structure, perhaps performing a comparison to provide a novelty point for this manuscript?
Line 59
Interesting results have been reported for chitosan (ref 22) and polysiloxanes (ref 23)……..”
On line 62, “This paper marks a continuation of studies into NAC grafted onto functional siloxane polymers”. How do results presented in the current study differ from ref 23 (published in 2018 by the same group)? It seems the main difference is bacteria strains been used: ref 23 used ATCC typed strains, and the current manuscript used wild isolates. Authors need to emphasize the unique part of the current study.
Line 110
In Figure 1, how does the amount of NAC (0.25% w/v) be determined? The amounts of NAC used in this study were not consistent, for examples, Figure 1 used 0.25%w/v; Figure 3 used 0.5% w/v; and Figure 4 used 0.05% w/v in NAC-Si-2 polymers due to the limit solubility. Please define the correct and reasonable numbers.
Line 123
Figure 1 seems not necessary. The inhibitory effect of NAC on bacteria had been reported before, and the effect of complex medium on NAC antimicrobial activity had also been reported (Ref 39). The recommendation for authors is to put more focus on antibiofilm activity of NAC, rather than antimicrobial activity.
Line 152-154
Authors confirmed the reported results (Ref 7). Please specify the new findings in the current manuscript.
Line 157-158
“Therefore, it was decided to evaluate the capacity of the tested bacteria to degrade NAC”.
Since the degradation of NAC by bacteria had been suggested, should it be better to conduct the degradation experiment first before testing it against varied bacteria strains?
Line 191-193
Please describes, or at least suggests the advantage(s) of using either ladder-like or linear-like polymer structure that NAC is grafted on? In addition, why were only two bacterial strains (A. hydrophila and C. freundi) tested, instead of all isolates?
Line 195-196
Figure 4 showed the effect of using different growth medium on NAC. How about using Si-1 and Si-2 only (no NAC grafted) as another control?
Line 203
Authors stated the incubation time was 48 h. How about 24 h? It would be better to have multiple time points instead of one time point.
Line 206-208
“These results show that the wild-bacterial isolates are more resistant to NAC………..described in our previous report (Ref 23)”.
This sentence is confusing. It is reasonable to assume varied bacteria strains possess different resistance to NAC and its derivatives. Authors need to clarify the point.
Author Response
Dear Reviewer 2,
We are very grateful for your thorough review of our paper, which has helped us to improve the quality of our publication. We have taken all your remarks into consideration, and have revised our paper in accordance with your comments.
We would like to take this opportunity to express our sincere thanks to Reviewer 2 who identified areas of our manuscript that needed corrections or modifications. We hope you will find our article after correction suitable for publication in International Journal of Molecular Sciences, section Molecular Microbiology.
Yours faithfully,
Anna Rygala & Dorota Kregiel

Round 2
Reviewer 2 Report
Authors have addressed questions properly. Thank you.